# Reconstructing 12-lead ECG from 3-lead ECG using variational autoencoder to improve cardiac disease detection of wearable ECG devices

Xinyan Guan[1], Haoyu Wang[1,2], Yongfan Lai[3,4], Jiarui Jin[1,3,4], Jun Li[1], Qinghao Zhao[5], Deyun Zhang [6], Shijia Geng[6], Guanyu Mu[7,8], Yiping Wang[9], Rui Wu[10], Shenda Hong [1,11,12,13,14]*

**1** National Institute of Health Data Science, Peking University, Beijing, China, **2** University of Chinese Academy of Sciences, Beijing, China, **3** State Key Laboratory of General Artificial Intelligence, Beijing, China, **4** School of Intelligence Science and Technology, Peking University, Beijing, China, **5** Department of Cardiology, Peking University People's Hospital, Beijing, China, **6** HeartVoice Medical Technology, Hefei, China, **7** Department of Echocardiography, The First Affiliated Hospital of USTC, University of Science and Technology of China, Hefei, China, **8** Department of Cardiology, The Second Hospital of Tianjin Medical University, Tianjin, China, **9** Department of Cardiology, Ma'anshan 17th Metallurgical Hospital, Ma'anshan, China, **10** Department of Cardio-Pulmonary Function, Fuwai Central China Cardiovascular Hospital, Zhengzhou, Henan, China, **11** Institute of Medical Technology, Peking University Health Science Center, Beijing, China, **12** Institute for Artificial Intelligence, Peking University, Beijing, China, **13** Department of Emergency Medicine, Peking University First Hospital, Beijing, China, **14** State Key Laboratory of Vascular Homeostasis and Remodeling, NHC Key Laboratory of Cardiovascular Molecular Biology and Regulatory Peptides, Peking University, Beijing, China

* hongshenda@pku.edu.cn

## Abstract

Twelve-lead electrocardiograms (ECGs) are the clinical gold standard for cardiac diagnosis, offering comprehensive spatial coverage of the heart necessary for detecting conditions such as myocardial infarction (MI). However, their lack of portability limits continuous and large-scale deployment. In contrast, three-lead ECG systems are widely used in wearable devices due to their simplicity and mobility, but they often fail to capture pathologies localized in unmeasured regions. To bridge this gap, we propose WearECG, a Variational Autoencoder (VAE) method that reconstructs 12-lead ECGs from three leads (II, V1, V5). Our model includes architectural improvements to better capture temporal and spatial dependencies in ECG signals. We evaluate generation quality using MSE, MAE, and Fréchet Inception Distance (FID), and assess clinical validity via a Turing test with expert cardiologists. To further validate the diagnostic utility, we fine-tune ECGFounder—a large-scale pretrained ECG model—on a multi-label classification task involving over 40 cardiac conditions, including 6 different myocardial infarction locations using both real and generated signals. Experiments on the MIMIC and PTB-XL datasets show that our method produces physiologically realistic and diagnostically informative signals, with robust performance in downstream tasks. This work demonstrates the potential of generative modeling for ECG reconstruction and its implications for scalable, low-cost cardiac screening.

**Data availability statement:** The code of our method and evaluations is publicly available at https://github.com/PKUDigitalHealth/WearECG-reconstruction. Data used in this study (MIMIC-ECG) are available under standard access agreements from PhysioNet.

**Funding:** This work was supported by the Beijing Municipal Science and Technology Commission (Grant No. Z251100000725008), National Natural Science Foundation of China (62102008, 62172018), CCF-Tencent Rhino-Bird Open Research Fund (CCF-Tencent RAGR20250108), CCF-Zhipu Large Model Innovation Fund (CCF-Zhipu202414), PKU-OPPO Fund (BO202301, BO202503), and the Research Project of Peking University in the State Key Laboratory of Vascular Homeostasis and Remodeling (2025-SKLVHR-YCTS-02). The funders had no role in study design, data collection and analysis, decision to publish, or preparation of the manuscript.

**Competing interests:** I have read the journal's policy and the authors of this manuscript have the following competing interests: Two co-authors, Shijia Geng and Deyun Zhang, are employees of HeartVoice Co., Ltd. (company information: https://www.heartvoice.com.cn/en/about.html). All other authors declare that they have no competing interests.

## Author summary

Twelve-lead electrocardiograms (ECGs) are a cornerstone of clinical cardiac diagnosis, but their use is largely restricted to clinical settings due to bulky hardware and limited portability. In contrast, wearable ECG devices typically record only a small number of leads, which constrains their ability to detect cardiac abnormalities that manifest outside the measured regions. In this study, we introduce WearECG, a generative framework that reconstructs full 12-lead ECG signals from only three commonly available leads (II, V1, and V5). Using a variational autoencoder architecture with enhanced temporal and spatial modeling, our method generates physiologically realistic ECG waveforms that closely resemble clinical recordings. We evaluate the quality of the reconstructed signals using both signal-level metrics and expert cardiologist assessments, and further examine their diagnostic value by fine-tuning a large pretrained ECG foundation model on real and generated data. Experiments on two large clinical ECG datasets demonstrate that the reconstructed signals preserve clinically meaningful information and support accurate downstream diagnosis across a wide range of cardiac conditions. Our results highlight the potential of generative modeling to extend the clinical utility of wearable ECG devices and enable scalable, low-cost cardiac monitoring beyond traditional healthcare environments.

## Introduction

As the leading global cause of mortality, cardiovascular disease (CVD) [1,2] necessitates reliable diagnostic tools, among which the 12-lead electrocardiogram (ECG)—performed over 300 million times annually—has become fundamental. The 12-lead ECG is considered the gold standard for non-invasive cardiac assessment owing to its ability to provide comprehensive electrical activity mapping of the heart. However, despite its diagnostic value, the standard 12-lead ECG lacks portability. It is not suitable for continuous, ambulatory monitoring. As a result, many cardiovascular events occur without timely detection or intervention, contributing to the high mortality associated with conditions such as myocardial infarction and arrhythmia. To address this gap, wearable ECG devices have emerged as a promising solution, enabling long-term cardiac monitoring in daily life and facilitating early detection of critical events. Accordingly, the development of user-centric devices for pervasive ECG signal acquisition has emerged as a central goal in both academic research and commercial innovation, encompassing patch-type systems [3–5], smartwatches [6–8], and armband-based solutions [9–11].

Given that portable devices typically capture only a few leads, researchers have spent the past thirty years exploring how to reconstruct the full 12-lead ECG by exploiting the inherent inter-lead correlations. While initial methods were predominantly based on linear transformations, the advent of artificial intelligence has paved the way for more advanced and effective reconstruction strategies.

Several prior studies have explored the reconstruction of standard 12-lead ECG signals using conventional techniques. Early approaches leveraged linear transformation matrices to capture inter-lead correlations [12,13], while others employed temporal modeling strategies [14,15]. However, many of these methods depend on patient-specific algorithms [16], which significantly limit their generalizability. Given that the relationships among ECG leads vary across individuals, such hand-crafted approaches often fail to capture the nonlinear and dynamic nature of inter-lead dependencies. These limitations have motivated researchers to explore data-driven alternatives capable of learning more lexible and generalizable mappings from incomplete inputs to full 12-lead signals.

Supervised deep learning models have gained increasing attention in this context, particularly due to their ability to leverage large-scale annotated ECG datasets. Nejedly et al. [17] introduced an ensemble learning framework that integrates residual convolutional neural networks (CNNs) with an attention mechanism, achieving strong classification performance. Building on this, Gundlapalle and Acharyya [18] combined CNNs and LSTM modules to model both spatial and temporal patterns in single-lead to 12-lead reconstruction. Garg et al. [19] proposed a modified Attention U-Net for reconstructing multiple leads from a single input, while Chen et al. [20] introduced a Multi-Channel Masked Autoencoder capable of generalizing across diverse input configurations. These approaches demonstrate the potential of discriminative models in ECG reconstruction, but often require abundant labeled data and still face limitations when input information is highly sparse.

To address these challenges, recent work has turned to generative models that can synthesize plausible ECG signals even under severely under-constrained input conditions. Lee et al. [21] proposed a conditional generative adversarial network (CGAN) aligned with R-peaks to reconstruct precordial leads (V1–V6) from limb-lead inputs, achieving high fidelity in terms of SSIM (≈ 0.92) and PRD (≈ 7.21%). Seo et al. [22] introduced a WGAN-based architecture featuring a U-Net generator and CNN discriminator to generate multiple leads from a single-lead input, evaluated using Fréchet Distance and MSE. More recently, Joo et al. [23] developed EKGAN, a dual-generator GAN framework with a 1D U-Net discriminator, which demonstrated superior performance in reconstructing full 12-lead ECGs from Lead I, as validated by both quantitative metrics and expert cardiologists. These generative approaches offer a promising direction for robust ECG reconstruction in low-resource or real-world deployment scenarios.

Single-lead ECG inputs are widely used in portable and home-monitoring devices but provide limited spatial information, which restricts their clinical interpretability, particularly for detecting regional abnormalities such as myocardial infarctions. While three-lead inputs capture orthogonal cardiac directions and enable more robust reconstruction, existing models like Mason et al.'s stacked ResNet [24] suffer from architectural complexity that may limit clinical applicability. Moreover, most current generative methods rely on GAN frameworks, which are notorious for training instability, mode collapse, and sensitivity to hyperparameters, making them challenging to optimize and deploy in real-world clinical settings.

To address these limitations, we explore generating 12-lead ECGs from three-lead inputs using a carefully designed VAE framework, incorporating multi-scale residual blocks, attention-enhanced bottlenecks, and a structured latent distribution. An overview of the proposed framework is shown in Fig 1. The encoder-decoder architecture leverages downsampling with residual convolutional blocks and group normalization, while KL-regularized latent sampling ensures stable training. The learned latent space supports physiologically plausible reconstructions, even under severely limited input leads. Therefore, the contributions in this study are as follows:

- We develop a VAE-based model for reconstructing 12-lead ECGs from three commonly used leads: II, V1, and V5.

- Our model achieves high-fidelity reconstruction, with an overall MSE of 0.00100, MAE of 0.01782, and FID of 12.64 on the test set.

- Beyond signal-level evaluation, we assess the clinical validity of the reconstructed ECGs on multiple diagnostic classification tasks (e.g., acute MI/STEMI, atrial fibrillation, atrial lutter, left axis deviation), achieving consistently high AUROC scores.

PLOS Digital Health

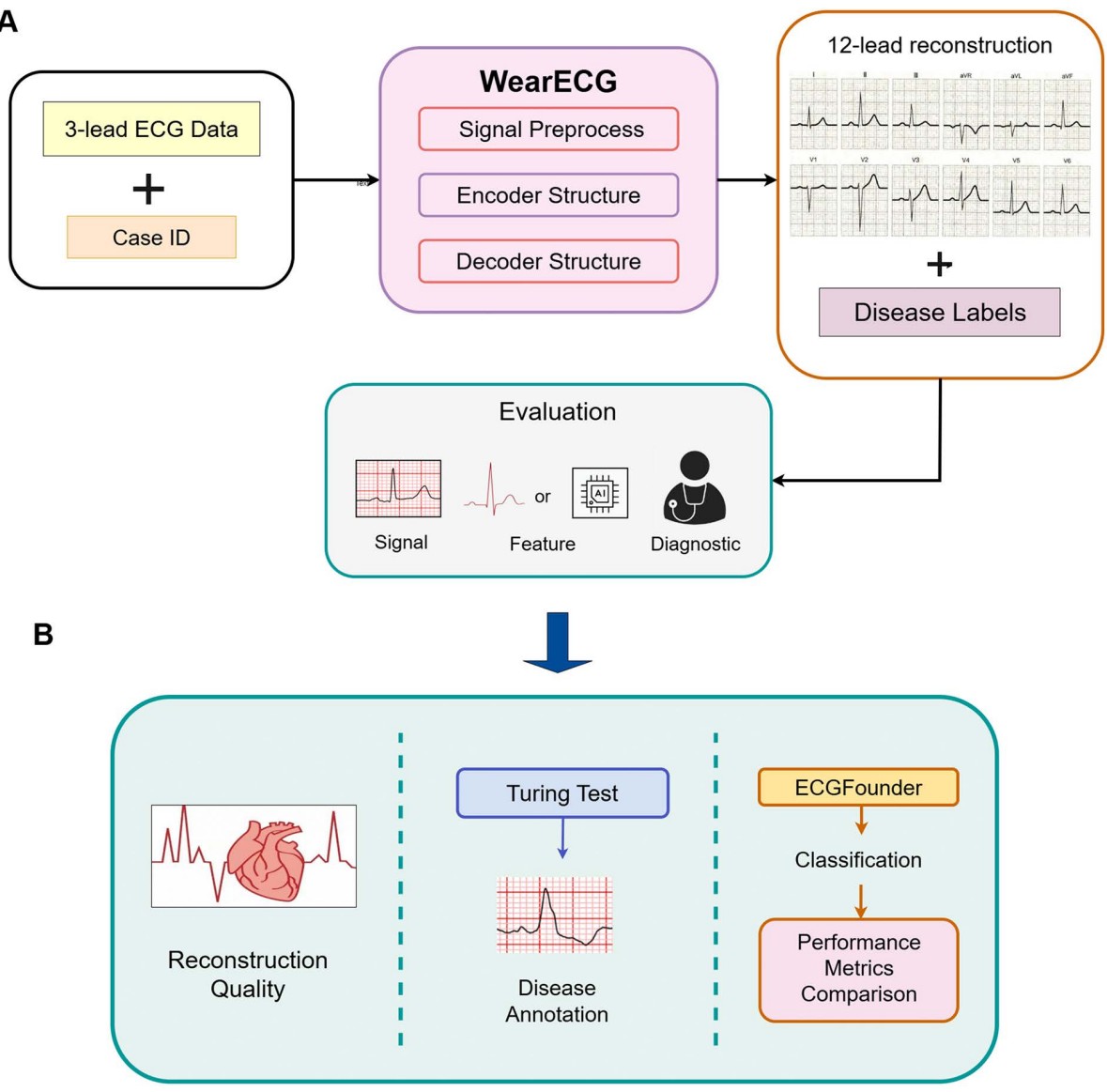

**Fig 1. Overview of our framework.** (A) Three-lead ECG signals are fed into a generative model to reconstruct full 12-lead ECGs. (B) The reconstructed ECGs are evaluated via multiple strategies, including signal-level metrics, fine-tuned disease classification using ECGFounder and cardiologist-involved Turing test.

## Results

To comprehensively evaluate the performance of our proposed model, we conducted a multi-level assessment spanning signal-level reconstruction quality, feature-level realism, and diagnostic utility. At the signal level, we quantitatively measured the similarity between generated and reference ECGs using metrics such as Mean Squared Error (MSE), Mean Absolute Error (MAE), and Fréchet Inception Distance (FID), capturing both numerical accuracy and perceptual closeness. To evaluate feature-level fidelity, we performed a blinded Turing test with three cardiology experts, who was tasked with distinguishing real from synthetic ECG signals, providing insight into the clinical plausibility of our generated data. Finally, at the diagnostic level, we fine-tuned a classification head on top of a frozen ECGFounder [25] backbone using the generated ECGs and evaluated performance across multiple cardiac conditions by AUROC, relecting how well the

synthetic signals preserve disease-specific characteristics. This comprehensive evaluation framework ensures a holistic understanding of the model's ability to generate ECGs that are accurate, realistic, and diagnostically meaningful.

## Method overview

We utilized the MIMIC-IV-ECG matched subset to train and validate our ECG generation model. This subset comprises approximately 800,000 ten-second 12-lead diagnostic ECG recordings from approximately 160,000 unique patients, collected at Beth Israel Deaconess Medical Center between 2008 and 2019. Each ECG is sampled at 500 Hz, stored in standard WFDB format, and is linked to patient metadata (demographics, RR interval) via shared subject_id and study_id with the MIMIC-IV Clinical Database [26].

When available, machine-generated summary measurements (e.g., RR intervals, QRS onset/end) and de-identified cardiologist text reports are provided for the same recordings. Identifiers and timestamps are privacy-protected using HIPAA-compliant de-identification, with shifted but internally consistent date-times enabling time-alignment across ECGs and other clinical events [26]. Because nearly all diagnostic ECGs for MIMIC-IV clinical patients are included, this dataset allows us to link ECG waveforms to hospitalization data such as admission, discharge diagnoses, age, sex, and clinical outcomes [26]. We retrieved age and sex information via the Clinical Database and calculated heart rate from RR interval records in the ECG measurements.

In the reconstruction task, we selected leads II, V1, and V5 because their spatial positions provide approximately orthogonal projections of cardiac electrical activity, forming a quasi-orthogonal triad that captures multidimensional cardiac vectors from different anatomical planes. Lead II primarily relects the inferior aspect, V1 focuses on the right ventricular and septal region, and V5 captures the lateral wall of the left ventricle. This complementary spatial coverage enables more accurate and comprehensive reconstruction of global cardiac electrical activity.

This lead selection is supported by classical vectorcardiography theory, which represents cardiac electrical activity in three approximately orthogonal dimensions. The Frank lead system and related vectorcardiographic models have demonstrated that a small set of quasi-orthogonal limb and precordial leads can capture most clinically relevant spatial information of cardiac electrical vectors [27,28]. Such theoretical foundations support the use of complementary lead combinations for effective ECG reconstruction.

Importantly, the simultaneous acquisition of leads II, V1, and V5 has already been demonstrated in existing ambulatory and long-term ECG monitoring systems based on chest-mounted patch and multi-electrode wearable designs. Such systems employ modified lead configurations (e.g., Mason–Likar–type arrangements) to enable stable multi-lead recording under realistic ambulatory conditions.

In particular, several medical-grade and research-grade wearable platforms have adopted electrode layouts that effectively approximate inferior, septal, and lateral cardiac views corresponding to leads II, V1, and V5, providing practical evidence that this configuration is technically feasible in real-world deployments (see S2 Fig for an illustrative example and technical details). Therefore, the proposed framework is compatible with current patch-based and garment integrated ECG acquisition systems and does not rely on unrealistic sensing assumptions. Moreover, ongoing advances in lexible electronics, textile-based electrodes, and low-power biosignal acquisition hardware are expected to further facilitate the integration of multi-lead sensing into compact wearable form factors.

We applied an improved Variational Autoencoder (VAE) architecture tailored for ECG signal characteristics to reconstruct the missing leads. The enhanced model architecture facilitates robust feature extraction and high-quality signal generation while preserving key diagnostic patterns. The reconstructed 12-lead ECGs were then used as input for the ECGFounder [25] model in downstream disease classification tasks, achieving high AUROC scores and demonstrating that the synthetic signals retain clinically relevant information.

The diagram below (Fig 2) illustrates the spatial relationship among these three leads, highlighting their complementary orientations in the frontal and horizontal planes, which is consistent with classical vectorcardiography theory.

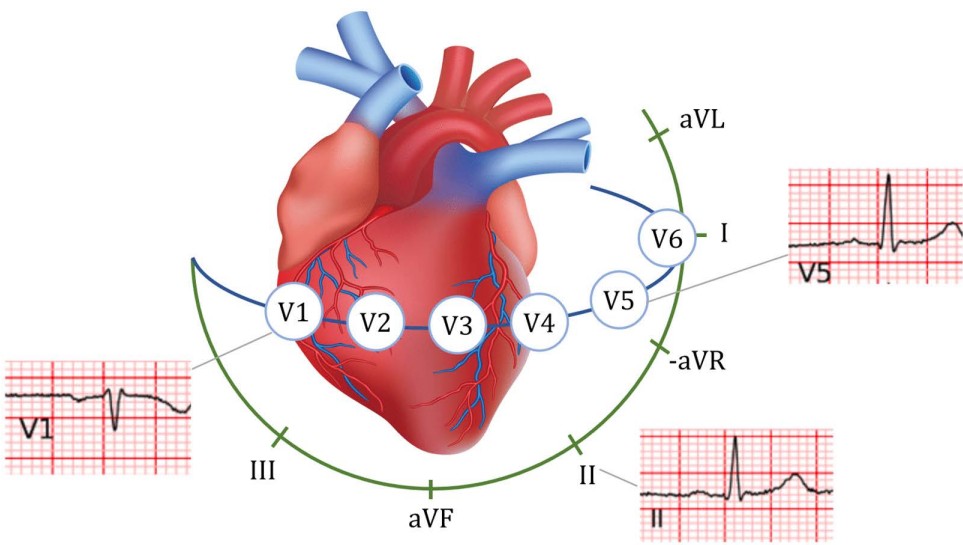

**Fig 2. Spatial relationship among leads II V1 V5.**

## WearECG can generate high fidelity ECG

To quantitatively evaluate the fidelity of generated ECG signals, we adopt three objective signal-level metrics: Mean Squared Error (MSE), Mean Absolute Error (MAE), and Fréchet Inception Distance (FID).

MSE and MAE measure point-wise reconstruction errors in the time domain, relecting waveform-level similarity between real and generated ECG signals. Let $\mathbf{r}_{ECG} \in R^{T \times C}$ and $\mathbf{g}_{ECG} \in R^{T \times C}$ denote the real and generated ECG signals, respectively, where T is the number of time points and C is the number of channels.

The MAE and MSE are defined as:

$$\text{MAE}\left(\mathbf{r}_{ECG},\ \mathbf{g}_{ECG}\right) = \frac{1}{T \cdot C}\ \sum_{t=1}^{T}\sum_{c=1}^{C}\left|r_{t,c} - g_{t,c}\right| \tag{1}$$

$$\text{MSE}\left(\mathbf{r}_{ECG},\ \mathbf{g}_{ECG}\right) = \frac{1}{T \cdot C}\ \sum_{t=1}^{T}\sum_{c=1}^{C}\left(r_{t,c} - g_{t,c}\right)^2 \tag{2}$$

MAE captures the average magnitude of reconstruction error, while MSE penalizes larger deviations more heavily. Lower values indicate more accurate signal-level reconstructions. Although FID was originally proposed for image evaluation, in this study it is computed based on high-level ECG representations extracted by ECGFounder [25], a large-scale ECG foundation model trained on over 10 million clinically annotated ECG recordings.

ECGFounder is supervised by cardiologist-labeled diagnostic categories and optimized to capture clinically meaningful patterns, including QRS morphology, ST-segment deviations, T-wave abnormalities, conduction disorders, and rhythm characteristics. It has also demonstrated strong generalization ability on reduced-lead ECGs in mobile and remote monitoring scenarios.

By operating in this learned feature space, FID measures the distributional similarity between generated and real ECG signals in terms of medically relevant waveform and temporal properties, rather than raw point-wise differences.

The FID is calculated as:

$$\text{FID} = \|\mu_r - \mu_g\|_2^2 + \text{Tr}\left(\Sigma_r + \Sigma_g - 2\left(\Sigma_r\Sigma_g\right)^{1/2}\right)$$

(3)

where $\mu_r$, $\Sigma_r$ and $\mu_g$, $\Sigma_g$ denote the mean and covariance matrices of feature embeddings for real and generated samples, respectively.

For each experimental setup, Table 1 list the MAE and MSE values per lead alongside the overall averaged scores.

In the main experiment using leads II, V1, and V5 as input, the per-lead MAE and MSE values are consistently low, with the MSE ranging from approximately 0.00061 to 0.00140. The overall MSE and MAE are 0.00100 and 0.01782, respectively. The corresponding FID score of 12.64 indicates that the reconstructed signals closely resemble real ECGs in the learned feature space. Clinically, this lead combination offers broad spatial coverage across the frontal and horizontal planes, effectively capturing both limb and precordial information.

To validate robustness, we further evaluated two alternative configurations: leads I, II, and V3, and the single-lead I setup. Both setups yield strong performance, with overall MSE/MAE scores of 0.00113/0.01856 and 0.00112/0.01863, respectively. These results highlight that while the proposed framework is robust to variations in input leads, carefully selecting leads with high 24 diagnostic value further enhances reconstruction accuracy and practical applicability.

## WearECG can enhance the performance of AI-ECG diagnosis

To rigorously evaluate the diagnostic capacity of the generated 12-lead ECG signals, we performed a downstream multi-label classification task using ECGFounder [25], a leading foundation model pretrained on over 10 million clinical ECG records. This task directly probes whether the generated signals retain sufficient pathological information to support real-world clinical decision making. Specifically, the model was tasked with predicting approximately 40 cardiovascular conditions—including arrhythmias, conduction abnormalities, myocardial infarction, and chamber enlargements—using multi-hot encoded labels that relect the frequent co-occurrence of cardiac diseases. This setting imposes a high bar for diagnostic fidelity, as successful classification requires the presence of nuanced, disease-specific signal features.

To isolate the diagnostic value of the generated signals, we froze the pretrained ECGFounder encoder and trained a lightweight classification head solely on synthetic ECGs. Prior to training, all signals were normalized on a per-lead basis via Z-score standardization. Performance was comprehensively evaluated using AUROC, sensitivity, specificity, and optimal threshold per condition, providing insights into both global and condition-specific diagnostic utility. Strong performance in this task indicates that our generation framework does more than mimic surface waveform morphology—it embeds clinically meaningful patterns that generalize to real-world diagnostic objectives, underscoring the potential value of synthetic ECGs in training, benchmarking, and augmenting downstream clinical models.

Using the MIMIC dataset for downstream evaluation (Table 2), the synthetic ECGs generated by our WearECG model achieved a macro-average AUROC of **0.8333**, closely approaching the original 12-lead configuration's performance (0.8465) and substantially outperforming both the 3-lead (0.7837) and 1-lead (0.7545) setups. For critical cardiac conditions such as sinus bradycardia (AUROC = 0.9882), atrial fibrillation (AUROC = 0.9751), and premature ventricular complexes (AUROC = 0.9551), our model maintained high sensitivity and specificity, indicating effective preservation of key diagnostic features in the synthetic signals.

To assess the generalizability of our model across datasets, we additionally evaluated downstream classification performance on the PTB-XL dataset (Table 3). Consistent with the MIMIC results, WearECG-generated synthetic ECGs maintained competitive AUROC scores, demonstrating effective transfer of learned representations to a distinct clinical dataset.

Importantly, we further evaluated WearECG on regional myocardial infarction (MI) classification across six anatomical locations (anterior, anterolateral, anteroseptal, inferior, lateral, and septal). As shown in Table 4, our model consistently achieved high AUROC scores for all regions (macroaverage AUROC = **0.8764**), demonstrating its ability to preserve

**Table 1. Signal-level evaluation metrics for all experiments on the MIMIC dataset with different input leads. FID is computed for overall reconstructed signals.**

| Lead Name Score | Lead Index | MSE | MAE | FID |
|---|---|---|---|---|
| **Main experiment (Input leads: II, V1, V5)** | | | | |
| I | 0 | 0.00075 | 0.01691 | |
| III | 2 | 0.00087 | 0.01833 | |
| aVR | 3 | 0.00061 | 0.01425 | |
| aVL | 4 | 0.00067 | 0.01570 | |
| aVF | 5 | 0.00073 | 0.01568 | |
| V2 | 7 | 0.00126 | 0.02051 | |
| V3 | 8 | 0.00135 | 0.01984 | |
| V4 | 9 | 0.00138 | 0.01997 | |
| V6 | 11 | 0.00140 | 0.01916 | |
| Overall | Metrics | 0.00100 | 0.01782 | 12.64 |
| **Comparative experiment (Input leads: I, II, V3)** | | | | |
| I | 0 | 0.00085 | 0.01826 | |
| III | 2 | 0.00061 | 0.01468 | |
| aVR | 3 | 0.00067 | 0.01571 | |
| aVL | 4 | 0.00070 | 0.01578 | |
| aVF | 5 | 0.00162 | 0.02073 | |
| V2 | 7 | 0.00125 | 0.02047 | |
| V3 | 8 | 0.00148 | 0.02071 | |
| V4 | 9 | 0.00150 | 0.02088 | |
| V6 | 11 | 0.00149 | 0.01981 | |
| Overall | Metrics | 0.00113 | 0.01856 | 11.58 |
| **Comparative experiment (Input lead: I)** | | | | |
| II | 1 | 0.00076 | 0.01826 | |
| III | 2 | 0.00081 | 0.01565 | |
| aVR | 3 | 0.00067 | 0.01572 | |
| aVL | 4 | 0.00083 | 0.01575 | |
| aVF | 5 | 0.00182 | 0.02174 | |
| V1 | 6 | 0.00147 | 0.02348 | |
| V2 | 7 | 0.00143 | 0.02048 | |
| V3 | 8 | 0.00158 | 0.02171 | |
| V4 | 9 | 0.00172 | 0.02289 | |
| V5 | 10 | 0.00159 | 0.01880 | |
| V6 | 11 | 0.00172 | 0.01980 | |
| Overall | Metrics | 0.00131 | 0.01939 | 12.52 |

spatially localized infarct patterns. Detailed regional MI classification performance on the PTB-XL dataset is presented in Table 5. This indicates that WearECG not only captures overall cardiac abnormalities but also retains the diagnostic fidelity necessary for identifying specific infarct locations, which is critical for accurate clinical assessment and intervention planning.

In contrast, the 1-lead approach exhibited significantly degraded performance across most pathologies and infarct regions, highlighting its limited clinical utility. The 3-lead system, while better than 1-lead, still falls short of matching the fidelity of our reconstructed signals. These results emphasize the added value of our method in bridging the gap between

**Table 2. AUROC comparison across different ECG lead configurations on MIMIC Dataset: WearECG (ours), original 12-lead, 3-lead only, and 1-lead only.**

| Disease | WearECG (ours) | original 12-lead | 3-lead only | 1-lead only |
|---|---|---|---|---|
| NORMAL | 0.7958 | 0.8468 | 0.7753 | 0.8071 |
| SINUS RHYTHM | 0.9247 | 0.9393 | 0.9262 | 0.9397 |
| SINUS BRADY | 0.9882 | 0.9881 | 0.9878 | 0.9848 |
| AFIB | 0.9751 | 0.9773 | 0.9718 | 0.9700 |
| SINUS TACHY | 0.9910 | 0.9928 | 0.9928 | 0.9914 |
| LAXIS DEV | 0.8609 | 0.8562 | 0.9269 | 0.2827 |
| PVC | 0.9551 | 0.9391 | 0.9389 | 0.8539 |
| RBBB | 0.9411 | 0.9588 | 0.9367 | 0.8632 |
| LAE | 0.7290 | 0.8103 | 0.6391 | 0.6578 |
| PAC | 0.9722 | 0.9570 | 0.9530 | 0.9573 |
| PSVC | 0.8785 | 0.8852 | 0.8707 | 0.8614 |
| LBBB | 0.7519 | 0.7448 | 0.7873 | 0.7224 |
| LVH | 0.6881 | 0.7666 | 0.7129 | 0.5812 |
| QT SHORT | 0.7406 | 0.7741 | 0.6956 | 0.5987 |
| QT LONG | 0.8025 | 0.8224 | 0.8076 | 0.7977 |
| AFLUTTER | 0.9088 | 0.9068 | 0.8962 | 0.8785 |
| SINUS ARR | 0.9418 | 0.9533 | 0.9278 | 0.8961 |
| LAFB | 0.9065 | 0.9031 | 0.9603 | 0.5420 |
| RAXIS DEV | 0.8588 | 0.8540 | 0.7259 | 0.5496 |
| ECTOPIC ATR | 0.7621 | 0.8204 | 0.7989 | 0.8932 |
| SHORT PR | 0.7600 | 0.8216 | 0.7818 | 0.7248 |
| REPOL ABN | 0.6019 | 0.6534 | 0.5176 | 0.5189 |
| RAE | 0.6219 | 0.7086 | 0.7660 | 0.8165 |
| VOLT CRIT LVH | 0.6310 | 0.6159 | 0.7123 | 0.5627 |
| LPFB | 0.7822 | 0.8294 | 0.7759 | 0.3841 |
| 1ST AVB | 0.8752 | 0.9282 | 0.7953 | 0.8537 |
| RVH | 0.8047 | 0.7927 | 0.7599 | 0.6064 |
| ACUTE MI/STEMI | 0.8427 | 0.8734 | 0.6794 | 0.6870 |
| SVT | 0.8374 | 0.8540 | 0.8130 | 0.8265 |
| VT | 0.8815 | 0.8925 | 0.8854 | 0.8740 |
| EARLY REP | 0.7927 | 0.7954 | 0.6801 | 0.7243 |
| WPW | 0.8831 | 0.9050 | 0.8306 | 0.8558 |
| ACUTE | 0.8469 | 0.8774 | 0.7280 | 0.6922 |
| ACUTE MI | 0.8437 | 0.8720 | 0.6971 | 0.7213 |
| SVC | 0.8785 | 0.8853 | 0.8718 | 0.8620 |
| 2:1 AV COND | 0.7428 | 0.6904 | 0.6936 | 0.6256 |
| **Macro-AUROC** | **0.8333** | **0.8465** | **0.7837** | **0.7545** |

minimal lead configurations and full 12-lead ECGs, making WearECG a promising tool for practical and accurate cardiac monitoring in resource-constrained settings.

Importantly, the successful transfer of knowledge from ECGFounder to synthetic ECGs underscores the generalizability and clinical value of our generation framework. This finding highlights the potential utility of synthetic ECGs in model

**Table 3. Regional MI classification on MIMIC Dataset (AUROC).**

| Region | WearECG | 1-lead only | 3-lead only | Original 12-lead |
|---|---|---|---|---|
| Anterior | 0.8683 | 0.7966 | 0.8397 | 0.8728 |
| Anterolateral | 0.8880 | 0.7979 | 0.8297 | 0.8894 |
| Anteroseptal | 0.9378 | 0.7676 | 0.8624 | 0.9410 |
| Inferior | 0.8067 | 0.8180 | 0.8055 | 0.8117 |
| Lateral | 0.8782 | 0.7989 | 0.8182 | 0.8823 |
| Septal | 0.8794 | 0.6763 | 0.7839 | 0.8927 |
| **Macro-AUROC** | **0.8764** | **0.7759** | **0.8233** | **0.8817** |

**Table 4. AUROC comparison across different ECG lead configurations on PTB-XL Dataset: WearECG (ours) and original 12-lead.**

| Disease | WearECG (ours) | original 12-lead | 3-lead only |
|---|---|---|---|
| NORMAL | 0.8183 | 0.8468 | 0.8027 |
| SINUS RHYTHM | 0.8436 | 0.9393 | 0.7981 |
| SINUS BRADY | 0.9301 | 0.9881 | 0.9040 |
| AFIB | 0.9883 | 0.9770 | 0.9917 |
| SINUS TACHY | 0.9907 | 0.9928 | 0.9844 |
| LAXIS DEV | 0.8311 | 0.8562 | 0.8102 |
| PVC | 0.9772 | 0.9391 | 0.9751 |
| RBBB | 0.9076 | 0.9588 | 0.8905 |
| LAE | 0.6621 | 0.8103 | 0.6848 |
| PAC | 0.9532 | 0.9570 | 0.8786 |
| PSVC | 0.8765 | 0.8852 | 0.8456 |
| LBBB | 0.9801 | 0.9848 | 0.8814 |
| LVH | 0.7283 | 0.7666 | 0.6673 |
| QT LONG | 0.8680 | 0.8224 | 0.8862 |
| AFLUTTER | 0.9589 | 0.9068 | 0.9906 |
| LAFB | 0.8384 | 0.9031 | 0.8152 |
| ECTOPIC ATR | 0.8337 | 0.8204 | 0.8170 |
| RAE | 0.7511 | 0.8086 | 0.7243 |
| LPFB | 0.6734 | 0.8294 | 0.7857 |
| 1ST AVB | 0.6857 | 0.9282 | 0.8375 |
| RVH | 0.8475 | 0.8927 | 0.8383 |
| ACUTE MI/STEMI | 0.6473 | 0.8734 | 0.5287 |
| SVT | 0.8929 | 0.9540 | 0.8237 |
| VT | 0.8641 | 0.8925 | 0.8264 |
| Macro-AUROC | 0.8470 | 0.8931 | 0.8328 |

pretraining, algorithm benchmarking, and data augmentation, especially in privacy-sensitive or low-resource settings where real ECG data are scarce or difficult to share.

## WearECG can improve the ECG diagnosis of cardiologists

Moreover, we conducted a blinded Turing test involving three board-certified cardiologists to evaluate the feature-level fidelity of the generated electrocardiogram (ECG) signals. We constructed a balanced set of 50 ECG samples—consisting

**Table 5. Regional MI classification on PTB-XL Dataset(AUROC).**

| Region | WearECG | Original 12-lead | 3-lead only |
|---|---|---|---|
| Anterior | 0.5828 | 0.6465 | 0.6287 |
| Anterolateral | 0.9112 | 0.9431 | 0.7473 |
| Anteroseptal | 0.8756 | 0.9440 | 0.8400 |
| Inferior | 0.7701 | 0.8615 | 0.7374 |
| Lateral | 0.8460 | 0.9134 | 0.5594 |
| Septal | 0.7447 | 0.9464 | 0.6748 |
| **Macro-AUROC** | **0.7885** | **0.8758** | **0.6979** |

of both real and generated signals—which was randomly shufled and presented under strictly blind conditions. The task required each expert to independently classify every sample as either "real" or "generated" based solely on waveform morphology and clinical plausibility. The resulting classification accuracies were 52%, 44%, and 44%, respectively, which are statistically close to random guessing (i.e., 50%). This suggests that the generated signals successfully mimic key morphological and temporal patterns found in authentic ECGs, such as waveform shapes, lead-to-lead coherence, and rhythm characteristics. In the evaluation of model-assisted myocardial infarction (MI) recognition, we analyzed the same set of generated ECG samples from two complementary perspectives: a quantitative assessment using a confusion matrix and a qualitative examination through case studies.In this setting, the model provides reconstructed ECG signals as visual diagnostic references, while the final classification is performed by physicians.

For the confusion matrix evaluation (Fig 3), the model-generated ECG samples, including both MI and normal cases, were independently assessed by physicians for MI detection and localization. The final diagnostic decisions were made solely by physicians based on the reconstructed ECG signals, without access to the original recordings or any algorithmic predictions.

Therefore, all reported performance metrics in this section relect physician diagnostic performance when interpreting the generated ECGs, rather than the classification performance of any automated algorithm.

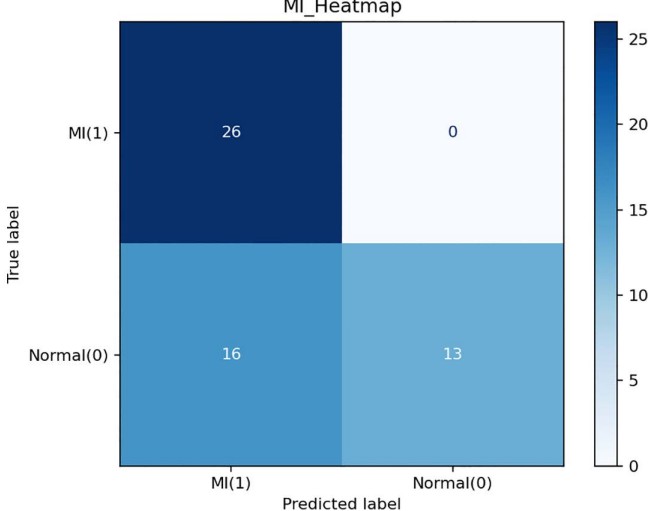

**Fig 3. Confusion matrix for model-assisted myocardial infarction (MI) detection.**

PLOS Digital Health

The evaluation yielded 26 true positives, 13 true negatives, 16 false positives, and 0 false negatives across 55 samples, corresponding to an accuracy of 70.9%, sensitivity of 100%, specificity of 44.8%, precision of 61.9%, and an F1-score of 76.5%.

Notably, the results demonstrate perfect sensitivity but relatively low specificity, indicating a conservative diagnostic tendency toward over-identifying potential MI cases. This behavior relects a preference for minimizing false negatives at the expense of increased false positives, which is often desirable in clinical screening scenarios to avoid missing critical conditions. This performance pattern can be attributed to multiple factors. First, physicians were instructed to make decisions solely based on reconstructed ECGs, which may contain subtle reconstruction uncertainty or mildly exaggerated abnormal patterns, potentially increasing false-positive judgments. Second, given the screening-oriented evaluation setting, physicians may have intentionally adopted a cautious diagnostic strategy when faced with ambiguous cases. In addition, the relatively limited sample size in this experiment may amplify variability in specificity estimation. Nevertheless, excessive false positives may increase clinical workload, highlighting the need for further optimization of reconstruction quality, decision thresholds, and calibration strategies in future work.

In the case study analysis, we selected representative samples to evaluate the effectiveness of model-generated ECGs in assisting physicians with myocardial infarction (MI) localization. Accurate localization of infarct regions is critical for guiding timely and appropriate clinical interventions, including treatment planning and risk assessment.

These results in Table 6 indicate that the model-generated ECGs closely align with physician assessments, effectively assisting in MI localization and providing useful reference for clinical interpretation. By accurately reconstructing ECG signals, the model offers a valuable tool for detecting and localizing myocardial infarction, which is critical for timely clinical decision-making, risk assessment, and treatment planning. This demonstrates that reconstructed ECGs can play an important role in supporting cardiologists in diagnosing heart conditions, particularly in identifying infarct regions, and may help reduce the likelihood of misdiagnosis in clinical practice. Additional cases are provided in the supplementary for further reference.

## Discussion

In this study, we propose a generative framework for reconstructing standard 12-lead ECG signals from only three input leads and demonstrate its effectiveness through both signal-level and diagnostic-level evaluations. The strong consistency between reconstructed and original ECGs in waveform similarity, feature representations, and downstream myocardial infarction classification indicates that the generated signals preserve clinically meaningful physiological information.

The confusion matrix analysis shows high sensitivity but relatively low specificity, relecting a conservative diagnostic tendency toward minimizing false negatives. Such behavior is often desirable in clinical screening scenarios to avoid missing critical cardiac events, suggesting that the proposed system may serve as a supportive triage and screening tool, while final diagnosis should remain under physician supervision.

Although the proposed framework utilizes leads II, V1, and V5 due to their complementary spatial information, simultaneously acquiring these leads using current consumer-grade wearable devices remains technically challenging. Most existing wearable systems are limited to simplified lead configurations, which may restrict immediate deployment in ambulatory settings. Nevertheless, ongoing advances in lexible electronics and multi-electrode sensing platforms are expected to enable more practical multi-lead acquisition in the future, providing opportunities for real-world integration of the proposed approach.

**Table 6. Representative case study samples for model-assisted MI localization.**

| Physician 1 | Physician 2 | Physician 3 | Model Prediction |
|---|---|---|---|
| Septal | Anteroseptal | Anteroseptal | Anteroseptal |
| Inferior/Septal | Inferior | Inferior | Inferior |

Several limitations should be acknowledged. The current evaluation is mainly based on specific public datasets, and generalizability to broader populations and device types requires further validation. In addition, ECG signals collected in clinical databases are generally cleaner than those acquired in real-world wearable environments, which are often affected by motion artifacts, electrode displacement, and ambient noise. The domain gap between hospital-grade recordings and ambulatory wearable signals remains a practical challenge for real-world deployment. Moreover, the framework relies solely on waveform data and does not incorporate multimodal clinical information, such as textual reports or medical records. While ECGFounder provides an objective evaluation backbone, it cannot fully replace expert cardiologist judgment. Future work will focus on large-scale external validation, multimodal integration, extension to additional cardiac conditions, and incorporation of physiological priors to further enhance robustness, interpretability, and clinical applicability.

## Methods

### Problem definition

The electrocardiogram (ECG) is a non-invasive tool that records the heart's electrical activity over time, characterized by waveforms such as the P-wave, QRS complex, and T-wave, which correspond to distinct phases of the cardiac cycle. The standard 12-lead ECG, derived from 10 electrodes placed on the limbs and chest (as detailed in Table 7), provides a spatially comprehensive view of cardiac function and is widely used for diagnosing various cardiovascular conditions.

We formulate 12-lead ECG reconstruction as a generative task, where the input contains only partial information from three selected leads. Given a segment $x \in R^{L \times T}$ representing an ECG with $L = 12$ leads over $T$ time steps (e.g., $T = 1000$ for 2 seconds at 500Hz), we simulate 3-lead input by masking out 9 leads:

$$X_{masked} = X \odot (1 - M) \tag{4}$$

Here, $M \in \{0, 1\}^{L \times T}$ is a binary mask with zeros at the retained leads II, V1, V5 and ones elsewhere. The model is trained to reconstruct the full 12-lead ECG $y \in R^{12 \times T}$ from $x_{masked}$. We adopt a standard Variational Autoencoder (VAE) framework. The encoder maps the masked ECG to a latent distribution $q(z|x_{masked})$, from which we sample a latent vector z. The decoder then reconstructs the full ECG $\hat{y} = f_\theta (z)$. The model is optimized using a combination of:

• Mean squared error between $\hat{y}$ and $y$

• KL divergence between the approximate posterior and prior

• Optional perceptual loss based on a pretrained ECG encoder

**Table 7. Electrode configuration in the standard 12-lead ECG system.**

| Lead | Electrode Position |
|------|-------------------|
| I | Left Arm, Right Arm |
| II | Left Foot, Right Arm |
| III | Left Foot, Left Arm |
| aVR | Right Arm |
| aVL | Left Arm |
| aVF | Left Foot |
| V1 | 4th intercostal space, right sternal border |
| V2 | 4th intercostal space, left sternal border |
| V3 | Midpoint between V2 and V4 |
| V4 | 5th intercostal space, midclavicular line |
| V5 | Lateral to V4, left midaxillary line |
| V6 | Lateral to V5, left midaxillary line |

To assess robustness and clinical applicability, we apply comparative experiments with multiple lead configurations, including:

- Single-lead input (e.g., lead I only)

- Three-lead input (e.g., I–II–V3 and II–V1–V5)

This setup enables the model to learn to infer missing leads from partial observations, mimicking clinical scenarios where limited-lead monitoring is available. What's more, it relects realistic scenarios in low-resource or wearable monitoring environments.

### WearECG overview

We propose a two-stage framework to reconstruct 12-lead ECG signals from 3-lead inputs and enable multi-label disease classification; we illustrate the representation in Fig 3. The first stage adopts an enhanced Variational Autoencoder (VAE) architecture [29], which extends the basic VAE structure by introducing several critical improvements for ECG signal modeling: (1) deep residual convolutional blocks that capture richer temporal representations; (2) multi-head self-attention modules to model long-range dependencies between leads; and (3) a hierarchical encoder-decoder design with progressive downsampling and upsampling to aggregate global context efficiently. This symmetric encoder-decoder backbone leverages structured latent representations through reparameterization to generate full-lead ECGs from partial inputs. Additionally, a perceptual loss is optionally employed during training to better preserve clinically meaningful features in the reconstructed signals, enhancing their diagnostic value.

In the second stage, we fine-tune ECGFounder [25], a pre-trained ECG classification model, by attaching a lightweight classification head for downstream diagnosis tasks such as myocardial infarction, conduction blocks, and arrhythmias, formulated as multi-label prediction. The downstream classification framework is illustrated in Fig 4. The entire pipeline is modular and can be trained end-to-end or in a decoupled manner, demonstrating robustness and clinical relevance across various lead configurations.

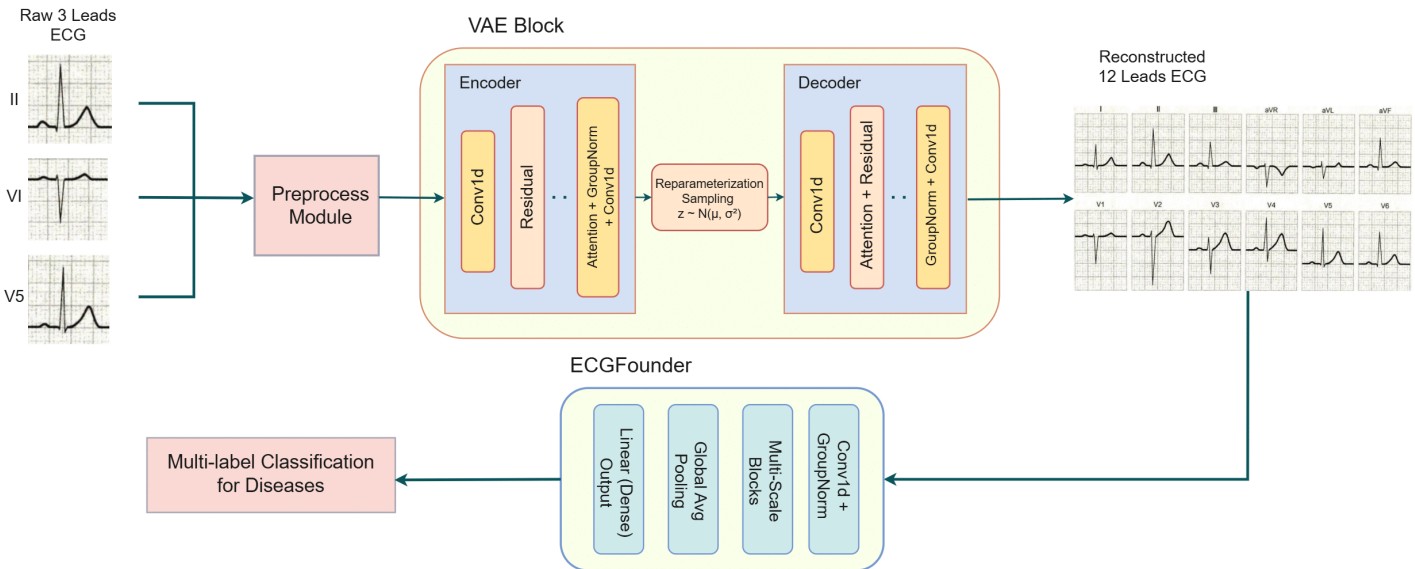

**Fig 4. Framework overview.**

## Dataset and data preprocessing

The raw 12-lead ECG signals underwent a multi-step preprocessing pipeline to ensure data quality and uniformity for model training. First, the leads were reordered to a standardized sequence to maintain consistency across datasets by mapping their original order to a target reference layout. Next, all signals were resampled to a fixed frequency of 500 Hz using Fourier-based interpolation, which preserved signal integrity while adjusting the time series length. Missing values (NaNs) within the signals were replaced by averaging neighboring data points within a defined window, effectively mitigating missing data without causing significant bias.

Although optional in this study, per-lead Z-score normalization was applied to align data distributions across samples and aid model convergence. To handle the large dataset efficiently, records were processed in batches, each undergoing NaN replacement, resampling, and optional normalization before saving. Finally, to prevent data leakage, train-test splits were created by grouping records according to *subject_id*, ensuring that all records from a given subject appear exclusively in either the training or testing set.

## WearECG Encoder: Learning latent structures from limited leads

The encoder is constructed using a multi-layer one-dimensional convolutional neural network (1D CNN) tailored to process 12-lead electrocardiogram (ECG) signals. The input signal $\mathbf{x} \in \mathbb{R}^{T \times 12}$, where T is the temporal length, is initially projected to a higher dimensional feature space through a convolutional layer, increasing channels from 12 to 128. The network then applies residual blocks and downsampling layers to extract hierarchical features, reducing temporal resolution while increasing channel depth to 512, resulting in a compressed representation $\mathbf{h} \in \mathbb{R}^{T' \times C}$ with $T' < T$.

Two separate convolutional layers predict the parameters of the approximate posterior distribution over the latent space:

$$\mu = f_\mu(\mathbf{h}), \quad \log\sigma^2 = f_\sigma(\mathbf{h}), \tag{5}$$

where $\mu$, $\log\sigma^2 \in \mathbb{R}^{T' \times d}$, and d is the latent dimension (here d = 4).

The latent variable $\mathbf{z}$ is then sampled using the reparameterization trick:

$$\mathbf{z} = \mu + \sigma \odot \epsilon; \quad \epsilon \sim N(\mathbf{0}, \mathbf{I}), \tag{6}$$

where $\odot$ denotes element-wise multiplication, ensuring differentiability of the sampling process for backpropagation.

## WearECG Decoder: Reconstructing twelve-lead signals from latent space

The decoder receives the latent representation $\mathbf{z} \in \mathbb{R}^{T' \times d}$ and aims to reconstruct the original ECG signal $\hat{\mathbf{x}} \in \mathbb{R}^{T \times 12}$. It employs a series of residual blocks integrated with multi-head self-attention mechanisms, which model long-range temporal dependencies. The decoder progressively upsamples the temporal dimension from T' back to T while reducing the channel depth from d back to 12.

Formally, the decoding process can be described as:

$$\hat{\mathbf{x}} = g_e(\mathbf{z}), \tag{7}$$

where $g_\theta(\cdot)$ represents the decoder network parameterized by $\theta$, incorporating convolutional, residual, attention, and upsampling layers.

The model is trained under the variational autoencoder framework by optimizing the evidence lower bound (ELBO):

$$L(\theta, \phi; \mathbf{x}) = E_{q_\varphi(\mathbf{z}|\mathbf{x})}\left[\log p_\theta(\mathbf{x}|\mathbf{z})\right] - KL\left(q_\varphi(\mathbf{z}|\mathbf{x}) \| p(\mathbf{z})\right), \tag{8}$$

where

$$q_\phi \left( \mathbf{z} | \mathbf{x} \right) = N \left( \mathbf{z}; \mu; \mathrm{diag} \left( \sigma^2 \right) \right), \ p \left( \mathbf{z} \right) = N \left( \mathbf{0}, \ \mathbf{I} \right), \tag{9}$$

with $q\phi$ the encoder's approximate posterior and $p(\mathbf{z})$ the prior.

The reconstruction loss is computed as the mean squared error (MSE):

$$\mathcal{L}_{\mathrm{recon}} = \frac{1}{N} \sum_{i=1}^{N} \| \mathbf{x}_i - \hat{\mathbf{x}}_i \|^2, \tag{10}$$

and the KL divergence term has a closed-form expression:

$$KL = -\frac{1}{2} \sum_{j=1}^{d} \left( 1 + \log \sigma_j^2 - \mu_j^2 - \sigma_j^2 \right). \tag{11}$$

The total loss is thus:

$$L = L_{\mathrm{recon}} + \beta \ KL, \tag{12}$$

where $\beta$ is a hyperparameter weighting the KL term (e.g., $10-4$).

Optimization is performed using the AdamW optimizer, with parameters updated by

$$\theta, \varphi \leftarrow \theta, \varphi - \eta \nabla_{\theta, \varphi} L, \tag{13}$$

where $\eta$ is the learning rate scheduled by a OneCycle learning rate policy to facilitate convergence.

To avoid data leakage, the dataset is split by subject according to their *subject_id*, ensuring 459 that all samples from a single subject belong exclusively to either the training or testing sets.

**Fine-tuned ECGFounder**

In the downstream classification stage, we fine-tune ECGFounder, a deep neural network pretrained on large-scale 12-lead ECG datasets. This study addresses a multi-label classification task on 12-lead ECG signals, targeting approximately 40 cardiac disease categories, including myocardial infarction [30], various conduction blocks [31], and arrhythmias. Labels are represented as multi-hot vectors, enabling simultaneous detection of multiple coexisting conditions. The task demands robust generalization and fine-grained feature extraction from the model.

ECGFounder employs a modular 1D convolutional architecture tailored for multi-lead ECG signals. The backbone network consists of multiple residual bottleneck blocks, each containing sequential $1 \times 1$, $k \times 1$, and $1 \times 1$ convolutions, combined with batch normalization and Swish activation functions. To maintain temporal resolution and reduce information loss, custom SAME padding convolution and max-pooling layers are applied. The network progressively downsamples the input signal through multiple stages, aggregating hierarchical features while incorporating squeeze-and-excitation modules for channel-wise attention.

Prior to model input, each ECG lead is standardized using Z-score normalization [32] to mitigate offset and amplitude scaling issues. During fine-tuning, the pretrained weights are loaded, and the backbone parameters are frozen to retain learned representations. A lightweight classification head—a fully connected linear layer—is appended to the extracted

deep features for multi-label prediction of cardiovascular conditions such as myocardial infarction, conduction blocks, and arrhythmias. Input ECG signals are standardized per lead by Z-score normalization before feeding into the model. Training uses a multi-label binary cross-entropy loss, optimized with Adam, with hyperparameters empirically selected for convergence.

The entire pipeline demonstrates robustness and clinical relevance across various lead configurations, enabling effective downstream diagnosis from reconstructed or raw ECG signals.

### Training setup

The model was trained on a single NVIDIA RTX 3090 GPU using the PyTorch framework. We utilized an efficient data loading pipeline that shufled the training data and processed it in batches of size 16. To optimize the model parameters, the AdamW optimizer was employed with an initial learning rate of $1 \times 10^{-5}$. A OneCycle learning rate scheduler was applied, with a maximum learning rate of $5 \times 10^{-5}$ and a warm-up period corresponding to 20% of the total training steps, which facilitated faster and more stable convergence.

Training was conducted for 10 epochs. The loss function combined a mean squared reconstruction loss and a Kullback–Leibler divergence (KLD) regularization term, where the KLD weight was set to $1 \times 10^{-4}$ to balance reconstruction fidelity and latent space regularization. To reduce GPU memory consumption and accelerate training without compromising performance, mixed precision training was enabled via PyTorch's Automatic Mixed Precision (AMP) feature.

To prevent data leakage and ensure the robustness of model evaluation, the dataset was split by *subject_id*, such that all samples belonging to the same subject were contained exclusively in either the training or validation sets. Data pre-processing, including normalization and consistent lead reordering, was performed during data loading to maintain input uniformity.

Model checkpoints were saved at the end of each epoch to enable recovery and facilitate model selection. Although an early stopping mechanism was configured, it was not triggered as training proceeded stably without overfitting.

### Supporting information

**S1 Table. Additional case study samples illustrating model-assisted myocardial infarction localization.** This table presents additional clinical cases used in the physician Turing test, comparing myocardial infarction localization results from multiple cardiologists and the proposed model to further assess model-assisted diagnostic consistency. (PDF)

**S1 Fig. Additional reconstruction examples of 12-lead ECGs generated from the II/V1/V5 input configuration.** Representative examples are shown to qualitatively demonstrate the fidelity of reconstructed waveforms across diverse ECG morphologies beyond those presented in the main text. (PDF)

**S2 Fig. Practical wearable electrode configuration enabling acquisition of II, V1, and V5 leads.** A representative chest-mounted, patch-based electrode layout is illustrated to demonstrate the feasibility of acquiring the proposed reduced-lead configuration in realistic ambulatory settings. (PDF)

### Author contributions

**Conceptualization:** Deyun Zhang, Shenda Hong.

**Data curation:** Xinyan Guan, Shijia Geng, Guanyu Mu, Yiping Wang, Rui Wu.

**Formal analysis:** Yongfan Lai, Jiarui Jin, Jun Li, Qinghao Zhao, Guanyu Mu, Yiping Wang, Rui Wu.

**Investigation:** Yongfan Lai, Shijia Geng.

**Methodology:** Xinyan Guan.

**Project administration:** Jun Li.

**Resources:** Deyun Zhang.

**Supervision:** Shenda Hong.

**Validation:** Xinyan Guan, Jun Li, Qinghao Zhao, Yiping Wang, Rui Wu.

**Visualization:** Xinyan Guan, Guanyu Mu.

**Writing – original draft:** Xinyan Guan.

**Writing – review & editing:** Xinyan Guan, Yongfan Lai, Jiarui Jin, Jun Li, Shenda Hong.

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
