## [Decision Letter · Decision Letter 0]

2 Feb 2026

PDIG-D-25-01249Reconstructing 12-Lead ECG from 3-Lead ECG using Variational Autoencoder to Improve Cardiac Disease Detection of Wearable ECG DevicesPLOS Digital Health Dear Dr. Zhang, Thank you for submitting your manuscript to PLOS Digital Health. After careful consideration, we feel that it has merit but does not fully meet PLOS Digital Health's publication criteria as it currently stands. Therefore, we invite you to submit a revised version of the manuscript that addresses the points raised during the review process. Please submit your revised manuscript by Mar 04 2026 11:59PM. If you will need more time than this to complete your revisions, please reply to this message or contact the journal office at digitalhealth@plos.org. Please include the following items when submitting your revised manuscript:* A letter that responds to each point raised by the editor and reviewer(s). You should upload this letter as a separate file labeled 'Response to Reviewers'. This file does not need to include responses to any formatting updates and technical items listed in the 'Journal Requirements' section below.* A marked-up copy of your manuscript that highlights changes made to the original version. You should upload this as a separate file labeled 'Revised Manuscript with Track Changes'.* An unmarked version of your revised paper without tracked changes. You should upload this as a separate file labeled 'Manuscript'. If you would like to make changes to your financial disclosure, competing interests statement, or data availability statement, please make these updates within the submission form at the time of resubmission. Guidelines for resubmitting your figure files are available below the reviewer comments at the end of this letter. We look forward to receiving your revised manuscript. Kind regards, Chenxi YangAcademic EditorPLOS Digital Health Nicole Li-JessenSection EditorPLOS Digital Health Leo Anthony CeliEditor-in-ChiefPLOS Digital Healthorcid.org/0000-0001-6712-6626  **Journal Requirements:**

i. Please clarify all sources of financial support for your study. List the grants, grant numbers, and organizations that funded your study, including funding received from your institution. Please note that suppliers of material support, including research materials, should be recognized in the Acknowledgements section rather than in the Financial Disclosure.

ii. State the initials, alongside each funding source, of each author to receive each grant. For example: "This work was supported by the National Institutes of Health (####### to AM; ###### to CJ) and the National Science Foundation (###### to AM)."

iii. State what role the funders took in the study. If the funders had no role in your study, please state: “The funders had no role in study design, data collection and analysis, decision to publish, or preparation of the manuscript.”

iv. If any authors received a salary from any of your funders, please state which authors and which funders.

2. We ask that a manuscript source file is provided at Revision. Please upload your manuscript file as a .doc, .docx, .rtf or .tex.

3. Please upload separate figure files in .tif or .eps format. Also, remove the figures from your manuscript file but keep the legends.

4. Please provide an Author Summary. This should appear in your manuscript between the Abstract (if applicable) and the Introduction, and should be 150–200 words long. The aim should be to make your findings accessible to a wide audience that includes both scientists and non-scientists. Sample summaries can be found on our website under Submission Guidelines:

https://journals.plos.org/digitalhealth/s/submission-guidelines#loc-parts-of-a-submission

5. We notice that your supplementary figures and table ‘Appendix A and B’ are included in the manuscript file. Please remove them and upload them with the file type 'Supporting Information'. Please ensure that each Supporting Information file has a legend listed in the manuscript after the references list.

6. Please note that your Data Availability Statement is currently missing the DOI/accession number of each dataset OR a direct link to access each database from PhysioNet. If your manuscript is accepted for publication, you will be asked to provide these details on a very short timeline. We therefore suggest that you provide this information now, though we will not hold up the peer review process if you are unable.

7. Some material included in your submission may be copyrighted. According to PLOS’s copyright policy, authors who use figures or other material (e.g., graphics, clipart, maps) from another author or copyright holder must demonstrate or obtain permission to publish this material under the Creative Commons Attribution 4.0 International (CC BY 4.0) License used by PLOS journals. Please closely review the details of PLOS’s copyright requirements here: PLOS Licenses and Copyright. If you need to request permissions from a copyright holder, you may use PLOS's Copyright Content Permission form.

Potential Copyright Issues:

a. Figures 1 and 2: Please confirm whether you drew the images / clip-art within the figure panels by hand. If you did not draw the images, please provide (a) a link to the source of the images or icons and their license / terms of use; or (b) written permission from the copyright holder to publish the images or icons under our CC-BY 4.0 license. Alternatively, you may replace the images with open source alternatives. See these open source resources you may use to replace images / clip-art:

- https://openclipart.org/

If the reviewer comments include a recommendation to cite specific previously published works, please review and evaluate these publications to determine whether they are relevant and should be cited. There is no requirement to cite these works unless the editor has indicated otherwise.  **Additional Editor Comments (if provided):****Reviewers' Comments:** Reviewer's Responses to Questions

**Comments to the Author**

1. Does this manuscript meet PLOS Digital Health’s publication criteria? Is the manuscript technically sound, and do the data support the conclusions? The manuscript must describe methodologically and ethically rigorous research with conclusions that are appropriately drawn based on the data presented.

Reviewer #1: Yes

Reviewer #2: Yes

Reviewer #3: Yes

2. Has the statistical analysis been performed appropriately and rigorously?

Reviewer #1: Yes

Reviewer #2: Yes

Reviewer #3: Yes

3. Have the authors made all data underlying the findings in their manuscript fully available (please refer to the Data Availability Statement at the start of the manuscript PDF file)?

Reviewer #1: Yes

Reviewer #2: Yes

Reviewer #3: Yes

4. Is the manuscript presented in an intelligible fashion and written in standard English?

Reviewer #1: Yes

Reviewer #2: Yes

Reviewer #3: Yes

5. Review Comments to the Author

Reviewer #1: This paper introduces a generative framework named WearECG, which aims to utilize Variational Autoencoder (VAE) technology to reconstruct clinical-standard 12-lead electrocardiograms (ECG) from the 3-lead (II, V1, V5) signals commonly used in wearable devices. Experimental results demonstrate that the signals reconstructed by this method on the MIMIC and PTB-XL datasets possess high physiological realism and diagnostic value, with a Macro-AUROC (0.8333) closely approaching the performance of the original 12-lead signals (0.8465).

Overall, the ideas proposed in this paper are novel and hold significant practical value, effectively resolving the contradiction between the lack of portability in standard 12-lead devices and the insufficient diagnostic information provided by single or three-lead devices. Furthermore, passing the Turing test by senior experts proves that the generated signals have strong physiological realism and clinical utility.

My revision suggestions are as follows:

- FID is typically used for image evaluation. While the authors explain that it is based on features extracted by ECGFounder, they should more elaborately explain why these high-level features were chosen to measure 1D signal distribution similarity and discuss the metric’s interpretability in the ECG domain.

- The authors selected leads II, V1, and V5 and explained their spatial orthogonality. I suggest citing more classic Vectorcardiography theories in the introduction or methods section to strengthen the argument for the optimality of this "three-lead combination" as input.

- In the description at line 170, it is stated that "overall MAE and MSE are 0.00100 and 0.01782." However, in Table 1 and the description at line 98, 0.00100 corresponds to MSE, while 0.01782 (or 0.01783) corresponds to MAE. Please rectify this discrepancy.

- The titles of Table 4 and Table 5 show inconsistent spelling between "PTBXL" and "PTB-XL." I recommend using "PTB-XL" uniformly throughout the paper.

- The manuscript interchangeably uses "Macro-AUC" (Tables 2, 3) and "Macro-AUROC" (body text). I suggest using Macro-AUROC consistently across all tables and text.

- There is an incomplete sentence at line 232 (referenced as line 243): "we . A balanced set..." Please revise this for clarity and grammatical correctness.

- In the text (around line 242/253), there is a reference to a "confusion matrix (Figure X)," but the corresponding figure is subsequently labeled as Figure 3. Please correct this placeholder in the document.

Reviewer #2: The paper is well-written, the methodology is sound, and the results demonstrate that the reconstructed signals retain significant diagnostic value, particularly for myocardial infarction localization. However, there are a few minor errors in the text and areas where the discussion could be strengthened before publication.My comments are in the attachment.

Reviewer #3: This manuscript proposes a deep learning framework named WearECG, which utilizes a Variational Autoencoder (VAE) to reconstruct standard 12-lead electrocardiograms (ECG) from a 3-lead subset (II, V1, V5). The study addresses a critical clinical need by bridging the gap between the portability of wearable devices and the diagnostic comprehensiveness of full 12-lead systems, The authors employ a multi-level evaluation strategy, including signal-level metrics (MSE, MAE, FID), a feature-level "Turing test" with cardiologists, and downstream diagnostic tasks using the ECGFounder foundation model. The experimental results demonstrate that the reconstructed signals achieve a Macro-AUC of 0.8333, closely approaching the 0.8465 achieved by original 12-lead signals, which indicates significant clinical potential.

Key Strengths

1. Comprehensive Evaluation Framework: The integration of signal fidelity, expert human assessment, and automated diagnostic validation (via a large-scale pretrained model) provides a robust evidence chain for the model’s utility.

2. Data Scale and Robustness: The use of the large-scale MIMIC-IV-ECG dataset for training and the PTB-XL dataset for cross-dataset validation ensures the statistical reliability and generalizability of the findings.

3. Architectural Innovation: The framework successfully incorporates multi-scale residual blocks and attention mechanisms to capture complex temporal and spatial dependencies within ECG signals.

Weaknesses:

1. Clinical Feasibility of Lead Selection: The study utilizes leads II, V1, and V5 due to their spatial orthogonality. However, in the context of wearable devices (e.g., patches or smartwatches), placing electrodes to capture these specific leads simultaneously can be challenging. The authors should add a brief discussion regarding the practical feasibility of this lead configuration in real-world ambulatory monitoring.

2. Analysis of Diagnostic Metrics: In the model-assisted MI detection task, the confusion matrix (Figure 3) shows a sensitivity of 100% but a relatively low specificity of 44.8%. Please provide an analysis of this result—specifically, whether the model/physician bias tends toward over-diagnosis to prevent missing critical cases.

3. Numerical Consistency: There is a discrepancy in the reported FID scores. The abstract/text mentions 12.64, while Table 1 lists it as 11.34. Why are they different?

6. PLOS authors have the option to publish the peer review history of their article (what does this mean?). If published, this will include your full peer review and any attached files.

**Do you want your identity to be public for this peer review?** For information about this choice, including consent withdrawal, please see our Privacy Policy.

Reviewer #1: No

Reviewer #2: No

Reviewer #3: No

**Figure resubmission:** While revising your submission, we strongly recommend that you use PLOS’s NAAS tool (https://ngplosjournals.pagemajik.ai/artanalysis) to test your figure files. NAAS can convert your figure files to the TIFF file type and meet basic requirements (such as print size, resolution), or provide you with a report on issues that do not meet our requirements and that NAAS cannot fix.

After uploading your figures to PLOS’s NAAS tool - https://ngplosjournals.pagemajik.ai/artanalysis, NAAS will process the files provided and display the results in the "Uploaded Files" section of the page as the processing is complete. If the uploaded figures meet our requirements (or NAAS is able to fix the files to meet our requirements), the figure will be marked as "fixed" above. If NAAS is unable to fix the files, a red "failed" label will appear above. When NAAS has confirmed that the figure files meet our requirements, please download the file via the download option, and include these NAAS processed figure files when submitting your revised manuscript. **Reproducibility:** To enhance the reproducibility of your results, we recommend that authors of applicable studies deposit laboratory protocols in protocols.io, where a protocol can be assigned its own identifier (DOI) such that it can be cited independently in the future. Additionally, PLOS ONE offers an option to publish peer-reviewed clinical study protocols. Read more information on sharing protocols at https://plos.org/protocols?utm_medium=editorial-email&utm_source=authorletters&utm_campaign=protocols

---

## [Decision Letter · Decision Letter 1]

16 Mar 2026

Reconstructing 12-Lead ECG from 3-Lead ECG using Variational Autoencoder to Improve Cardiac Disease Detection of Wearable ECG Devices

PDIG-D-25-01249R1

Dear Mr Zhang,

We are pleased to inform you that your manuscript 'Reconstructing 12-Lead ECG from 3-Lead ECG using Variational Autoencoder to Improve Cardiac Disease Detection of Wearable ECG Devices' has been provisionally accepted for publication in PLOS Digital Health.

Best regards,

Chenxi Yang, Ph.D.

Academic Editor

PLOS Digital Health

**Additional Editor Comments (if provided):**

**Reviewer Comments (if any, and for reference):**

Reviewer's Responses to Questions

**Comments to the Author**

1. If the authors have adequately addressed your comments raised in a previous round of review and you feel that this manuscript is now acceptable for publication, you may indicate that here to bypass the “Comments to the Author” section, enter your conflict of interest statement in the “Confidential to Editor” section, and submit your "Accept" recommendation.

Reviewer #1: All comments have been addressed

Reviewer #2: All comments have been addressed

Reviewer #3: All comments have been addressed

2. Does this manuscript meet PLOS Digital Health’s publication criteria? Is the manuscript technically sound, and do the data support the conclusions? The manuscript must describe methodologically and ethically rigorous research with conclusions that are appropriately drawn based on the data presented.

Reviewer #1: Yes

Reviewer #2: Yes

Reviewer #3: Yes

3. Has the statistical analysis been performed appropriately and rigorously?

Reviewer #1: Yes

Reviewer #2: Yes

Reviewer #3: Yes

4. Have the authors made all data underlying the findings in their manuscript fully available (please refer to the Data Availability Statement at the start of the manuscript PDF file)?

Reviewer #1: Yes

Reviewer #2: Yes

Reviewer #3: Yes

5. Is the manuscript presented in an intelligible fashion and written in standard English?

Reviewer #1: Yes

Reviewer #2: Yes

Reviewer #3: Yes

6. Review Comments to the Author

Reviewer #1: No

Reviewer #2: The concerns have been addressed.

Reviewer #3: Thanks for the authors' response. I don't have any further comments.

7. PLOS authors have the option to publish the peer review history of their article (what does this mean?). If published, this will include your full peer review and any attached files.

**Do you want your identity to be public for this peer review?** For information about this choice, including consent withdrawal, please see our Privacy Policy.

Reviewer #1: None

Reviewer #2: None

Reviewer #3: No
